# Validating Constructs of the Depression, Anxiety, and Stress Scale-21 and Exploring Health Indicators to Predict the Psychological Outcomes of Students Enrolled in the Pacific Islands Cohort of College Students

**DOI:** 10.3390/ijerph21040509

**Published:** 2024-04-19

**Authors:** Rebecca H. Kim, Yvette C. Paulino, Yoshito Kawabata

**Affiliations:** 1College of Natural and Applied Sciences, University of Guam, Mangilao, GU 96923, USA; 2School of Health, University of Guam, Mangilao, GU 96923, USA; paulinoy@triton.uog.edu; 3College of Liberal Arts and Social Sciences, University of Guam, Mangilao, GU 96923, USA; kawabatay@triton.uog.edu

**Keywords:** DASS-21, construct validation, health indicators, psychological outcomes, Guam college students

## Abstract

The Depression, Anxiety, and Stress Scale-21 (DASS-21) has been used in various countries to assess the mental states of individuals. The objectives of this study were to validate the DASS-21 for use in Guam, an island that endures a high burden of mental health challenges, such as suicide, and examine the predictive impact of selected health indicators on DASS-21 variables. Three years of data (2017–2019) were pooled from the Pacific Islands Cohort of College Students (PICCS) study conducted annually at the University of Guam. In total, 726 students were included in the secondary data analysis. MPlus statistical software was used to perform a confirmatory factor analysis (CFA) for the validation and structural equation modeling (SEM) for the predictive modeling. The results from the CFA suggested an acceptable model fit (RMSEA: 0.073, CFI: 0.901, TLI: 0.889, RMR: 0.044), while SEM suggested that sleep quality and physical activity were significant predictors of DASS-21 variables. Therefore, the DASS-21 is a valid instrument for measuring depression, anxiety, and stress among emerging adults in Guam.

## 1. Introduction

Mental illness is an insidious and globally prevalent disease, estimated to affect one in four people at some point in their lives [1]. Amongst the mental health disorders, depression (4.4% or 321.2 million people worldwide) and anxiety disorders (3.6% or 262.8 million people worldwide) are known to be two disorders of the highest prevalence. Mental health disorders not only impact individuals’ well-being but also have a considerable epidemiological and economic impact in that they are associated with disease burden estimates for all major categories of communicable and non-communicable diseases, outpacing cancers and heart disease, as well as injuries. Moreover, the global economy loses an estimated cost of USD one trillion (or 12 billion workdays) each year to lost productivity caused by depression and anxiety disorders [2,3]. At worst, depression and anxiety disorders, especially when unaddressed, increase the risk of suicide [4].

On the other hand, stress is a normal reaction to everyday pressures, but it becomes unhealthy when it starts to impact one’s daily life [5]. Stress may be manifested physiologically, for example, by “heart palpitations, sweating, dry mouth, shortness of breath, fidgeting, accelerated speech”, and psychologically, through the “augmentation of negative emotions (if already being experienced), and longer duration of stress fatigue” [6]. Stress, when unaddressed, may put an individual at higher risk of many health problems, including digestive issues, heart disease, muscle tension, and weight gain [7].

Currently, about one-seventeenth of the world’s population suffers from mental disorders. However, nearly two-thirds of these people never seek treatment even though treatments are available, mainly due to stigma, discrimination, and neglect [2]. However, given the harmful effects mental disorders and stress have on health and the economy and the lack of treatment sought out by individuals suffering from mental disorders, it is essential to understand the causes of these disorders better to counteract the stigma and discrimination associated with them. 

In the present day, the etiology of mental disorders is understood using both somatogenic and psychogenic theories, in what is known as the bio-psycho-social model. This model attests that while one may be born with a genetic predisposition for a certain psychological disorder, certain psychological stressors need to be present for one to develop the disorder [8]. Some risk factors identified for depression include genetic predisposition [9], gender and age [10], underlying illnesses [10,11], unhealthy lifestyles, such as smoking and alcohol consumption [12,13], insomnia [14], gene–environment interplay [15], maladaptive parenting [16,17,18], and cultural differences [19]. 

Some risk factors identified for anxiety disorders include heritable influences [20,21,22,23,24,25,26,27,28,29], behavioral conditioning (such as classical aversive conditioning) [30], pathways in the central and peripheral nervous system [31,32,33,34,35], cognitive processes [36,37,38], cultural differences [39,40,41], age, and gender [42,43,44,45,46]. Another critical risk factor for the emergence of anxiety and depression is stress [47]. The diathesis–stress model suggests that psychopathology results from the interaction of risk factors in combination with precipitating stressful circumstances [48]. Hence, stress is also an important construct in understanding the development of psychopathology. 

One of the ways clinicians and researchers can help identify and assess the mental states of individuals is by using a psychological tool with good validity and reliability. Some of the common psychological tools used worldwide include the Beck Depression Inventory, Beck Anxiety Inventory, and Depression, Anxiety, and Stress Scale-21 (DASS-21), and extensive research has been undertaken to validate the constructs and to test the reliability of the tools in various countries [49,50]. 

In Guam, the Pacific Islands Cohort of College Students (PICCS) study has utilized the Depression, Anxiety, and Stress Scale-21 (DASS-21) to assess the status of mental health for Guam’s college student population. However, the construct validity and reliability of this psychological tool have not been tested, nor was any literature found on the validation of psychometric instruments for the Guam population. The DASS-21 was chosen and utilized for its broad applicability and convenience, as it is in the public domain and is the short form of the DASS-42.

To further introduce the scale, the DASS-21 is a self-report scale developed in Australia by Lovibond and Lovibond, which was designed to measure “the negative emotional state of depression, anxiety, and stress” [51]. According to the developers, this scale is not only valuable for conventionally measuring these negative emotional states of depression, anxiety, and stress, but also in further distinguishing between features of depression, anxiety, and stress, as the DASS-21 includes different subscales to measure the three states of depression, anxiety, and stress separately [51]. Hence, the DASS may be a helpful tool for researchers and clinicians in further defining, understanding, and measuring the ubiquitous emotional disturbance’s nature, etiology, and mechanisms. 

Thus far, the DASS-21 has been translated and validated in several different countries with high internal consistency, such as northern Vietnam [52], Turkey [53,54], Greece [55], China [56,57], Brazil [58], Italy [59], and Bahasa Malaysia [60]. The DASS-21 has also been validated amongst different demographics, such as other racial groups in the United States [61], genders [62], both clinical [63] and non-clinical samples [64], and adolescents [65,66,67] with good reliability scores. Some studies that examined the factor loading of the subscales onto the latent structures of the DASS-21 found that some items had a better factor loading when characterized under another latent structure, attributing the changes as part of the variation of emotional experiences between different cultures [56,58]. Hence, given the lack of literature on psychometric assessment in Guam, the primary objective of this study was to validate the DASS-21 among Guam’s college student population (Figure 1). 

Moreover, many studies worldwide are starting to elucidate the association between physical and mental health [68,69]. However, to date, no interdisciplinary study has been conducted to understand the impact of physical health on mental health in Guam. Hence, as a secondary objective to reduce the knowledge gap between disciplines, some fundamental health indicators, such as diet, sleep, and physical activity, were selected from the PICCS study, and their association with depression, anxiety, and stress outcomes was tested for in Guam’s college student population.

In understanding the role of health factors as indicators, studies have shown that the quality of diet is associated with lower levels of symptoms of depression, anxiety, and stress [70,71,72,73,74], the quantity and quality of sleep are associated with lower levels of symptoms of depression, anxiety, and stress [75,76,77,78,79,80,81], and physical activity is associated with lower levels of symptoms of depression, anxiety, and stress [82,83,84,85,86,87]. Hence, as suggested by the literature, the hypothesized model for Objective 2 was created to test for the association of health indicators with psychological outcome variables. (Figure 2). 

## 2. Materials and Methods

### 2.1. Procedures

This study was a secondary data analysis of representative, cross-sectional data collected by in-person interviews of college students from the 2017–2019 Pacific Islands Cohort of College Students (PICCS) study at the University of Guam. The PICCS project was conducted by the students in the HS 416: Research in Health Sciences class.

### 2.2. Participants

The sample comprised approximately 10% of the University of Guam student population, males and females, ages 18–29 for each year. Informed consent was obtained before administering the surveys and questionnaires to the participants.

The total number of participants for this study was 726, which resulted from combining the available datasets between 2017 and 2019. This number is well above the minimum recommended sample size (200) for testing the theoretical model of confirmatory factor analysis [88]. Combining them was also possible, as these samples were independent between each year. As for the demographics of the total participants, the mean age was 21.7 ± 2.3, and the proportion of male to female was 45.3% male to 54.7% female. The participants were also ethnically diverse (Table 1).

### 2.3. Measures

The following were the questionnaires selected from the PICCS survey to assess the variables of interest:Diet: The diet questionnaire was adapted from the 2013 Behavioral Risk Factor Surveillance System (BRFSS) survey [89]. This survey is a U.S. national self-report survey that collects and provides prevalence data about health-related risk behaviors, chronic health conditions, and the use of preventive services. The diet questionnaire selected for this study from the PICCS focused on the frequency of consuming fruits, vegetables, and fast foods. The respondents indicated how often they consumed the selected food types, and the responses for fruits and vegetables were scored as Daily = 3, Weekly = 2, Monthly = 1, and Seldom = 0. The fast-food responses were categorized as Daily = 0, Weekly = 1, Monthly = 2, and Seldom = 3. The scores for the frequency of each food type were summed up and ranged from 0 to 9, where 9 indicated a high frequency of fruit and vegetable consumption and a low frequency of fast-food consumption. In contrast, 0 indicated a low frequency of fruit and vegetable consumption and a high frequency of fast-food consumption;Sleep: The quantity and quality of sleep were assessed by the questions adapted from the Pittsburgh Sleep Quality Index (PSQI) [90]. As for the quantity of sleep, the question asked was, “How many hours of actual sleep do you get at night?” The participants were asked to provide a numerical response, and if answered in ranges (e.g., 6–7 h), the average of the two numbers (e.g., 6.5 h) was taken. As for the quality of sleep, the question asked was, “How would you rate your sleep quality overall?” The participants were asked to choose one of the following four options: “Very good”, “Fairly good”, “Fairly bad”, or “Very bad”. Here, the component score for the responses was coded 0–3, where 0 indicated “Very bad”, and 3 “Very good”;Physical activity: The physical activity in this study was evaluated using the Physical Activity Rating Questionnaire (PAR-Q) [91]. The participants were asked to use a number 0–7 to best describe their general activity level for the previous month. Zero indicated “avoid physical assertion”, while seven indicated running over 10 miles per week or spending over 3 h per week participating in comparable physical activity;Depression, anxiety, and stress: The constructs of depression, anxiety, and stress were measured by the 21-item short-form of the Depression, Anxiety, and Stress Scale (DASS-21) [51]. Participants rated how often they experienced the symptoms associated with depression, anxiety, and stress. “Never” was scored 0, “Sometimes” was scored 1, “Often” was scored 2, and “Almost always” was scored 3. The scores were added and multiplied by 2 to calculate the final score;Covariates: some possible covariates suggested by the literature that were accounted for in this study were gender, BMI, ethnicity, SES, and the year of data collection.

### 2.4. Data Analysis

Statistical analyses were conducted using MPlus software version 8.4 [92] and IBM SPSS software version 26 [93]. 

#### 2.4.1. For Objective 1: Construct Validation and Reliability Test of DASS-21

The validity of the scale of depression, anxiety, and stress was tested first by examining the normality of the data measured by the values of skewness and kurtosis in the distributional indices for each DASS-21 item. The cut-off score used for skewness was less than 2, and for kurtosis was less than 7, as suggested by Cohen et al. [94]. 

After testing for normal distribution, the confirmatory factor analysis was performed on MPlus software, where each item of the Depression, Anxiety, and Stress Scale-21 (DASS-21) was loaded onto the latent variables of depression, anxiety, and stress, following the three-factor model proposed by Lovibond and Lovibond [51]. 

The model-fit information was obtained and assessed following Hu and Bentler’s guidelines [95], which suggested a cut-off score of 0.95 or greater for the Comparative Fit Index (CFI), 0.06 or less for Root Mean Square Error of Approximation (RMSEA), and 0.08 or less for Standardized Root Mean Square Residual (SRMR) for good fit and following Hooper et al. [96] for acceptable fit. 

Moreover, the internal consistency of the DASS-21 was tested by running a reliability test on SPSS, and Cronbach’s alpha was obtained for each dimension of the DASS-21 and the whole test [97].

#### 2.4.2. For Objective 2: Predicting Psychological Outcomes with Health-Related Variables

The predictive effects of the health-related variables on depression, anxiety, and stress were tested first by running the descriptive statistics of the predictors and covariates with SPSS. Here, the missing values and non-normality of some of the variables were detected and adjustments were made. 

Then, correlational analyses were conducted using MPlus for continuous variables and SPSS MANOVA for categorical variables to examine the relationships between the health and psychological variables. 

Next, structural equation modeling (SEM) was conducted using MPlus to assess the hypothesized model (Figure 2). From the SEM analysis, the model results were interpreted using two two-tailed *p*-values, and model-fit information was assessed following Hu and Bentler’s [95] and Hooper’s [96] guidelines.

#### 2.4.3. Adjustments for Missing Data

For handling missing data, studies have shown that when more than 10% of the cases included missing data, using statistical methods, such as maximum-likelihood estimation or multiple imputation techniques, was superior to the conventional methods of dropping cases with missing scores, under the assumption that the cases are missing at random (MAR) [98,99]. Hence, variables missing at random (e.g., diet and physical activity) were included in the analyses, applying the advanced statistical method of maximum-likelihood estimation and Monte Carlo integration. For this validation study, advanced methods (maximum-likelihood estimation with Monte Carlo integration—part of the syntax to read the missing values on MPlus) were applied in the analyses containing more than 10% missing data. 

#### 2.4.4. Adjustments for Non-Normal Data 

To handle non-normal data, outliers were removed, inapplicable variables were recoded as missing, and the statistical method of bias-corrected bootstrapping was applied to the analyses. Bootstrapping is a method of random sampling with replacements. This procedure can sample each value multiple times or not at all while maintaining the data structure. It is useful for estimating statistical parameters where data are non-normal [100]. For the analyses in this study, bootstrapping was set to 5000 resamples.

## 3. Results

### 3.1. For Objective 1: Construct Validation and Reliability Test of DASS-21

The normality test for the items of the DASS-21 indicated that the items were all normally distributed. The results from the confirmatory factor analysis of the DASS-21 showed an acceptable model fit for the hypothesized three-factor structure (RMSEA = 0.073, CFI = 0.901, TLI = 0.889, and SRMR = 0.044). The factor loadings for all items were above 0.5, except for item 2, which showed a weak factor loading (0.327), which meant that item 2 showed relatively little influence on the measurement of the construct of anxiety. To test if the model fit improved after the removal of item 2, a confirmatory factor analysis was conducted again without the item. However, the removal of item 2 did not result in significant improvement to the model fit (RMSEA = 0.076, CFI = 0.902, TLI = 0.889, and SRMR = 0.045) (Table 2; Figure 3).

As for the reliability and the internal consistency of the DASS-21, Cronbach’s alpha for depression was 0.883, anxiety was 0.804, stress was 0.846, and the total was 0.933. The internal consistency for measuring the construct of anxiety improved by 0.21 (Cronbach’s alpha = 0.825) after removing item 2. Overall, the model fit information from the confirmatory factor analysis indicated a good model fit and reliability score.

### 3.2. For Objective 2: Predicting Psychological Outcomes with Health-Related Variables

The results from the descriptive statistics of the predictors and covariates showed that the mode score for diet, which was a categorical variable, was five, which indicated the median frequency of fruit and vegetable consumption. The mode score for physical activity was one, which indicated that many (23.4%) of the participants did not participate regularly in exercise but walked for pleasure. The mode of sleep quality was two, indicating that more than half (62.1%) of the participants rated their sleep quality as “fairly good”. The average hours of sleep the participants got were 6.19 ± 1.4 h. As for the covariates, the average BMI was 27.32 ± 2.64 kg/m^2^, which indicated that participants on average were overweight. Other noteworthy observations for the dataset were the missing values detected for some of the variables. More specifically, 30.6% of data on diet and 19.4% of data on physical activity rating questionnaires were missing at random in the combined dataset.

The normality test for the predictors and covariates suggested that the continuous variables were all normally distributed (Table 3). Moreover, there were some significant results from the correlational analyses for the continuous predictors. The results showed that sleep quality was negatively correlated with depression (r = −0.215, *p* < 0.01), anxiety (r = −0.151, *p* < 0.01), and stress (r = −0.204, *p* < 0.01). Physical activity was negatively correlated with depression (r = −0.124, *p* < 0.01) and anxiety (r = −0.136, *p* < 0.01), and among the covariates, gender showed a significant correlation with anxiety (r = 0.148, *p* < 0.01) and stress (r = 0.190, *p* < 0.01) (Table 4). 

For the categorical covariates, the results from the MANOVA showed no statistically significant differences between socioeconomic status, ethnicity, and the year of on the observed dependent variables of depression, anxiety, and stress scores.

The model results from the structural equation modeling with all predictors showed that sleep quality had a significant direct negative effect on the level of depression (β = −0.208, *p* < 0.01), anxiety (β = −0.143, *p* < 0.01), and stress (β = −0.197, *p* < 0.01), and physical activity had a significant direct negative effect on the level of depression (β = −0.116, *p* < 0.05) and anxiety (β = 0.127, *p* < 0.01). Gender had a significant effect on the level of anxiety (β = 0.147, *p* < 0.01) and stress (β = 0.193, *p* < 0.01) (Table 5). The model fit information from the SEM indicated an acceptable model fit for the hypothesized model (RMSEA = 0.061, CFI = 0.893, TLI = 0.878, SRMR = 0.045) (Table 6). 

## 4. Discussion

### 4.1. Objective 1: Construct Validation and Reliability Test of DASS-21 

The results from the confirmatory factor analysis of the DASS-21 showed a good model fit and reliability score, which implied that the DASS-21 instrument is a valid and reliable tool for measuring the constructs of depression, anxiety, and stress for Guam’s emerging adults. However, amongst the items, item 2, which measured the construct of anxiety, showed weak factor loading (0.327), which meant that the item showed relatively little influence on the measurement of the construct. Although item 2 showed low factor loading, removing it did not significantly improve the validity and reliability of the model. Moreover, a factor loading of 0.3 or greater was considered to “load on a particular factor” [101]. Hence, it is safe to keep it as part of the questionnaire but score it with the notion that item 2 may have little influence on the symptom of anxiety. 

### 4.2. Objective 2: Predicting Psychological Outcomes with Health-Related Variables 

The results from the data analysis from Objective 2 suggested that individuals with poorer sleep quality predicted greater levels of depression, anxiety, and stress, and individuals with lower levels of physical activity predicted greater levels of depression and anxiety (Figure 4). Moreover, the findings amongst the covariates showed that females were more likely to experience higher levels of anxiety and stress than males. 

#### 4.2.1. Sleep Quality and Mental Health

The results from the correlational analysis showed that sleep quality and depression, anxiety, and stress were significantly correlated. Furthermore, the results from the SEM revealed that poor sleep quality predicted higher levels of depression, anxiety, and stress for the emerging adult population. This observation was consistent with a previous study conducted by Rezaei et al. [102], where poor sleep quality was also significantly associated with psychological symptoms of depression, anxiety, and stress. 

Looking further into the literature on the possible explanation for the linkage between sleep quality and psychological symptoms, the mechanisms mediating these links are poorly understood. A potential mechanism suggested by several studies was that poorer sleep may be impairing the brain’s function of emotion regulation, leading to increased psychological distress [103]. However, more study needs to be conducted to better understand the linkages. 

#### 4.2.2. Physical Activity and Mental Health

The results from SEM further showed that low physical activity predicted higher levels of depression and anxiety. These findings also supported the findings from a longitudinal study that examined the relationship between physical activity and mental health (depression, anxiety, and burnout) among Swedish healthcare workers, where physical activity had a direct negative correlation and association with depression and anxiety. The longitudinal component of the study brought additional insight, where changes in physical activity were associated with changes in depression, anxiety, and burnout across time [104]. 

Moreover, numerous studies have discussed the possible mechanisms of how physical activity improves mental health. Exercise has been shown to increase the release and synthesis of several neurotrophic factors, such as the brain-derived neurotrophic factor (BDNF) and the insulin-like growth factor (IGF-1), which are related to better cognitive functioning, neurogenesis, angiogenesis, and plasticity. BDNF was also shown to have antidepressant effects. In addition, exercise also increased several neurotransmitters, such as serotonin, dopamine, acetylcholine, and norepinephrine, changing the cortical/subcortical activity [105]. There is overwhelming evidence in the literature suggesting that exercise ensures successful brain functioning. However, the linkages between exercise and mental health are complex, and more research needs to be carried out to elucidate the mechanism. 

#### 4.2.3. Other Non-Significant Predictors

On the other hand, the predictors of diet and sleep quantity did not show significant, direct associations with the levels of depression, anxiety, and stress.

Some possible explanations for these results may be that, for diet, the questionnaires were scored to result in one continuous variable, which is the type of variable easily deciphered by the MPlus statistical software and is consistent with other predictors. However, the process of combining the three separate variables of the frequency of fruit, vegetable, and fast-food consumption into one continuous variable of ‘diet’ may have ambiguated the score interpretation of the variable, possibly leading to low reliability of the scores, and perhaps influencing the statistical results. Perhaps, keeping the three variables separate as categorical variables, along with conducting a reliability test, may result in a more precise score interpretation.

As for sleep quantity, the instrument was shown to have good validity and reliability yet did not show significant associations with the level of depression, anxiety, and stress. To understand the difference in the result of why sleep quality rather than sleep quantity was associated with psychological outcomes, studies conducted by Pilcher et al. [106] and Bassett et al. [107] were reviewed. First, Pilcher et al. [106] assessed whether sleep quantity or sleep quality better measured the health and well-being of 117 college students. Their results showed that sleep quality was better related to measures of health, well-being, and sleepiness than sleep quantity. These results were independent of the effects of sleep quality on sleep quantity. Next, Bassett et al. [107] further assessed the impact of sleep quantity and sleep quality on cortisol responses to acute psychosocial stress in 73 college students, and their results showed that the self-reported quantity of sleep did not appear to affect cortisol stress responses, while perceived sleep quality impacted cortisol stress responses in a gender-dependent manner. Bassett et al. [107] also suggested that the difference in emotion-regulation strategies, reappraisal, and suppression may influence sleep quality and contribute to the gender-dependent relationship between sleep quality and cortisol. Hence, as suggested by these findings, the difference in the result in this study may be explained by the presence and the absence of the appraisal, where sleep quality and not sleep quantity revealed one’s appraisal to an independent stressor, which was shown to have a direct, independent influence on the physiological and psychological stress response. 

#### 4.2.4. Other Factors

Other possible explanations for why these predictors and covariates did not show significant associations may also be that these predictors may have a significant nonlinear relationship, a bidirectional relationship, or other different paths of relationship untested by the data analyses. These specific structural-equation-modeling analyses performed for this study mainly tested for the correlation and multiple regression of the observed independent variables on latent dependent variables. Because the analyses were primarily focused on testing the structural relationship of the hypothesized model, other possible bi-directional or nonlinear relationships between variables may have needed to be included or submitted for examination. 

### 4.3. Limitations

Several factors limited this study. Firstly, with this study being a secondary-data analysis, there were some challenges in dealing with the missing values and limitations in understanding the details of the data-collection procedure and the intentions in choosing specific survey questions. This may have impacted the scope of the research questions possible to ask and on the provision of context to interpret the results. 

Secondly, the fit index criteria employed exhibited a high degree of leniency, as there has been a recent surge in the utilization of more stringent metrics, following Hu and Bentler’s guidelines [95]. Additionally, there may be a limit to the generalizability of the findings in terms of the possible covariates that were not accounted for, such as culture, other groups of age and gender, SES, and other social factors, including the effects of state dependency.

Lastly, with the study design being cross-sectional, there was a limitation in interpreting the study findings, as correlation does not imply causation. Although from this study, one may infer the predictive role of physical activity and sleep quality on depression, anxiety, and/or stress, this study is limited in understanding the interactions and causations between these factors. So far, other studies have reported the bidirectional nature of these factors. However, the linkages between the factors are complex and are still poorly understood. Hence, more research, in general, is needed to elucidate the mechanism. 

### 4.4. Suggestions for Future Studies 

To address these limitations, our recommendation for future studies is to design an experimental study, using measurements that accurately capture both the subjective and objective data of each variable. This study should include a temporal element to better define the causal relationship of the independent and dependent variables.

Furthermore, given that the fit-index criteria exhibited a high degree of leniency, and the TLI value of 0.889 would be considered unsatisfactory with more stringent metrics, it would be beneficial to consider alternative models such as bifactor or ESEM or BESEM estimating methods. Additionally, it may be helpful to employ an exploratory factor analysis (EFA) to investigate the items lacking in strength. 

Moreover, the participants were exclusively university students. Hence, more studies would be required in other population segments such as adolescents, older adults, and clinical populations with medical or psychological alterations to make a correct estimate of the dimension of the test. 

Guam is unique in its location, diverse population, and culture. Its culture is influenced by the history of colonization by Spain, the United States, and Japan, as well as the immigrants from Asia and other Micronesian islands. Hence, incorporating some social factors, such as acculturation, other age and gender groups, and ethnicities, may improve the generalizability of the findings. 

Other possible directions for future studies would be to investigate the bidirectional association between these physical and psychological health-related variables and to further expand on the research on integrating various disciplines of social, natural, and applied sciences to strengthen the bridge and broaden the understanding between them. 

## 5. Conclusions

Some general inferences one can make from this study are that the DASS-21 is a valid and reliable instrument in measuring the constructs of depression, anxiety, and stress for Guam’s emerging adults and that some subjective health variables of sleep quality and level of physical activity may, independently and cumulatively, predict the levels of depression, anxiety, and stress among Guam’s emerging adults. 

The findings from this study may also bring clinical and research benefits in that, as the DASS-21 was found to be a sound and reliable instrument to measure the levels of depression, anxiety, and stress for Guam’s college-student population, clinicians and researchers may confidently use the DASS-21 instrument to better understand the psychological distress of Guam’s college students. Additionally, current evidence may imply that by improving sleep quality and exercising regularly, Guam’s emerging adults may reap the benefits of physical and psychological health. These findings suggest a more holistic approach for clinicians, with equal emphasis on health-related factors, such as sleep quality and physical activity, in understanding and treating the client’s distress. 

Lastly, during the global pandemic, studies worldwide have shown that people’s quality of sleep declined [108,109], and their physical activity decreased, while sedentary behaviors increased [110]. Hence, these findings may also help raise awareness of the further impacts of the changes in sleep and exercise patterns on one’s mental health and encourage further investigation into the factors to provide more insights and find preventive measures to improve not only the physical health but also the psychological health of the general public. 

## Figures and Tables

**Figure 1 ijerph-21-00509-f001:**
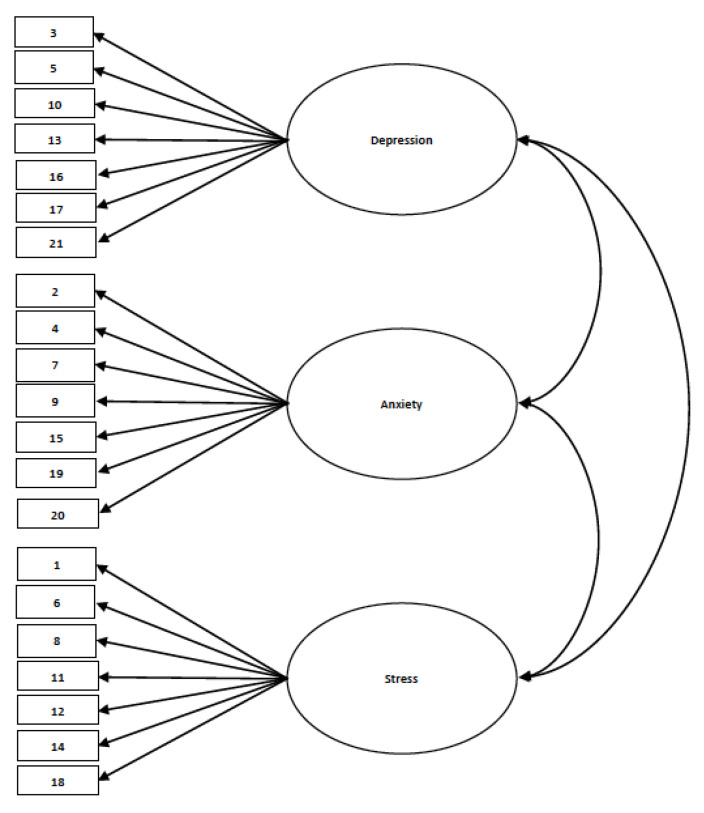
Hypothesized model for Objective 1. This figure shows the three-factor model of DASS-21 proposed by Lovibond and Lovibond [51]. (Generally, the ovals represent the latent constructs, and the rectangles represent the observed variables. The one-headed arrow indicates a linear regression, while the two-headed arrow indicates a correlation).

**Figure 2 ijerph-21-00509-f002:**
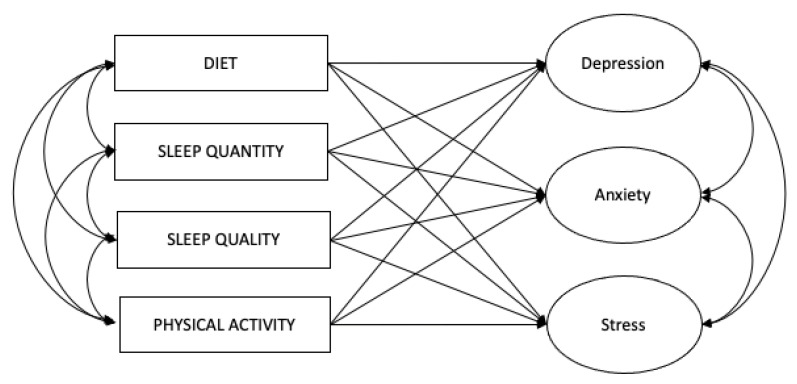
Hypothesized model for Objective 2. This figure is the hypothesized model of the relationship between the health-related predictor variables and psychological outcome variables.

**Figure 3 ijerph-21-00509-f003:**
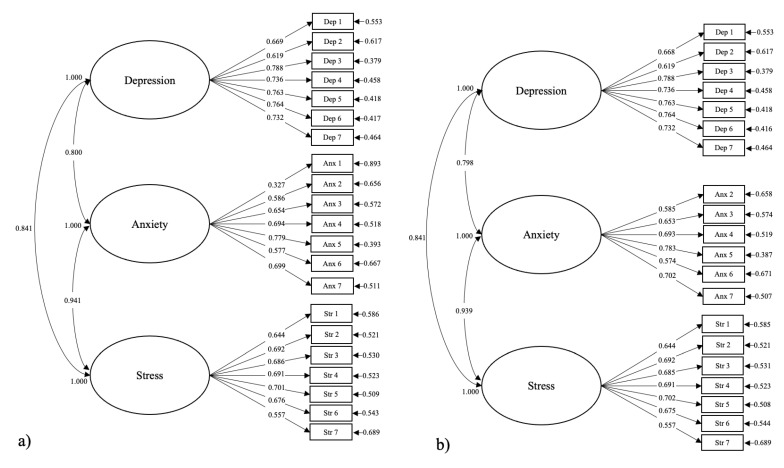
Results from confirmatory factor analysis. These figures are a diagram of the confirmatory factor analysis of the DASS-21 instrument with factor loadings (**a**) of all items and (**b**) without item 2.

**Figure 4 ijerph-21-00509-f004:**
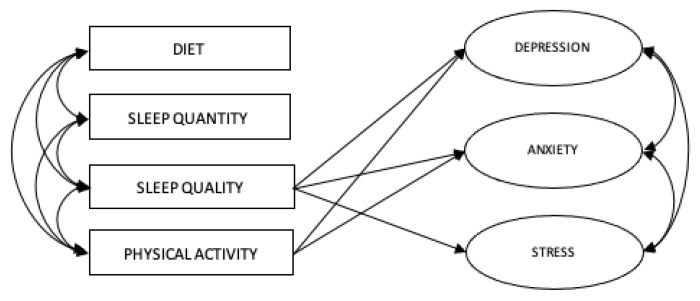
Structural-equation model with significant paths. (*n* = 726).

**Table 1 ijerph-21-00509-t001:** Demographics of participants by each year of data collection and by total.

Year		2017	2018	2019	Total
Participants		284	230	212	726
Average Age		21.73 ± 2.3	21.43 ± 2.04	22.06 ± 2.56	21.7 ± 2.3
Gender by Frequency (%)	Male:	*n* = 129, 45.4%	*n* = 109, 47.4%	*n* = 91, 42.9%	*n* = 329, 45.3%
	Female:	*n* = 155, 54.6%	*n* = 121, 54.6%	*n* = 121, 57.1%	*n* = 397, 54.7%
Ethnicities by Frequency (%)	Asian:	43.3%	43.5%	49.5%	45.2%
	Native Hawaiian/ Other Pacific Islander:	42.6%	46.5%	42.9%	43.9%
	White:	1.1%	1.7%	1.9%	1.5%
	American Indian/Alaskan Native:	0.4%	1.3%	0.5%	0.7%
	African American:	0.7%	0.0%	0.9%	0.6%
	Missing:	12%	7%	4.2%	8.1%

**Table 2 ijerph-21-00509-t002:** Model fit values for confirmatory factor analysis.

Model-Fit Indicators	Good Fit [95]	Acceptable Fit [96]	Model-Fit Values	
			(a) All Items	(b) Without Item 2
RMSEA	0.00 < x < 0.06	0.06 < x < 0.08	0.073	0.076
CFI	0.95 < x < 1	0.90 < x < 0.95	0.901	0.902
TLI	0.95 < x < 1	0.80 < x < 0.95	0.889	0.889
SRMR	x < 0.08	x < 0.08	0.044	0.045

RMSEA: Root Mean Square Error of Approximation; CFI: Comparative Fit Index; TLI: Tucker Lewis Index; SRMR: Standardized Root Mean Residual.

**Table 3 ijerph-21-00509-t003:** Normality test for continuous variables of predictors and covariates.

Variables	N	Min	Max	Mean	SD	Skewness	Kurtosis
DIET	504	0	9	5.42	1.47	−0.470	1.167
SLEEPQLT	723	0	3	1.71	0.68	−0.644	0.524
SLEEPQTY	719	1	13	6.19	1.40	0.129	1.727
PARQ	585	0	7	3.67	2.23	0.013	−1.336
AGE	726	18	29	21.73	2.33	0.620	0.278
GENDER	726	1	2	1.55	0.50	−0.189	−1.970
BMI	690	14.18	62.03	27.32	6.64	1.296	2.728

Gender: 1 = male, 2 = female.

**Table 4 ijerph-21-00509-t004:** Correlation analysis of the 2017–2019 dataset (*n* = 726, with missing values on diet and physical activity (w/ML, bootstrap (5000))).

Predictors and Covariates	Depression (Latent)	Anxiety (Latent)	Stress (Latent)
Diet	−0.017	0.008	−0.020
Sleep Quality	−0.215 **	−0.151 **	−0.204 **
Sleep Quantity	−0.013	−0.041	−0.070
PARQ	−0.124 **	−0.136 **	−0.054
Age	−0.058	−0.075	−0.057
Gender	0.041	0.148 **	0.190 **
BMI	0.029	0.007	−0.015

** *p* ≤ 0.01.

**Table 5 ijerph-21-00509-t005:** Model results for SEM using an advanced method with Monte Carlo, ML, BOOTSTRAP (5000), *n* = 726.

	Standardized Estimates (β)	Two-Tailed *p* Value
DEPRESSION ON		
DIET	−0.015	0.744
SLEEPQLT	**−0.208**	0.000
SLEEPQTY	−0.011	0.780
PARQ	**−0.116**	0.016
GENDER	0.039	0.349
ANXIETY ON		
DIET	0.015	0.764
SLEEPQLT	**−0.143**	0.002
SLEEPQTY	−0.039	0.405
PARQ	**−0.127**	0.007
GENDER	**0.147**	0.000
STRESS ON		
DIET	−0.015	0.743
SLEEPQLT	**−0.197**	0.000
SLEEPQTY	−0.067	0.112
PARQ	−0.043	0.355
GENDER	**0.193**	0.000

Bolded values mean *p* ≤ 0.05, SLEEPQLT: Sleep Quality, SLEEPQTY: Sleep Quantity, PARQ: Physical Activity Rating Questionnaire.

**Table 6 ijerph-21-00509-t006:** Model-fit values for structural equation modeling.

Model-Fit Indicators	Good Fit [95]	Acceptable Fit [96]	Model-Fit Values
RMSEA	0.00 < x < 0.06	0.06 < x < 0.08	0.061
CFI	0.95 < x < 1	0.90 < x < 0.95	0.893
TLI	0.95 < x < 1	0.80 < x < 0.95	0.878
SRMR	x < 0.08	x < 0.08	0.045

## Data Availability

The datasets analyzed during the current study and survey materials can be obtained by contacting Y.C.P. (paulinoy@triton.uog.edu).

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
