# Peer review of "Validating Constructs of the Depression, Anxiety, and Stress Scale-21 and Exploring Health Indicators to Predict the Psychological Outcomes of Students Enrolled in the Pacific Islands Cohort of College Students"

_ijerph, 2024, doi:10.3390/ijerph21040509_

Round 1

Reviewer 1 Report

Comments and Suggestions for Authors

 -In the introduction, it is essential to add reference studies that establish the presence and relationship of the variables studied, as well as the association of selected health indicators of diet, sleep, and physical activity with depression, anxiety, and stress.

-Add a relevant documentary research, which demonstrates the state of the art of the problem raised in this report.

-It is suggested to integrate the exploratory analysis of data, as a background to the confirmatory analysis of the same.

-It should be considered that the participants are exclusively university students, so more studies are required in other population segments such as adolescents, older adults and clinical population with medical or psychological alterations to make a correct estimate of the dimensions of the test.

Reviewer 2 Report

Comments and Suggestions for Authors

In summary, this research offers a well-crafted and enlightening addition to the examination of the psychometric characteristics of the DASS-21 among college students in the Pacific Islands Cohort. The study's objectives are well articulated, and the study's methodology and data analysis demonstrate a high level of rigor and suitability. This study provides a significant scholarly addition by demonstrating the utility of the questionnaire as a measurement instrument across many cultural contexts.

The primary remarks and critiques I have regarding the paper are as follows:

Incorporating a brief section in the introduction explaining the psychometric qualities of instruments across various countries, such as factor structures, would be beneficial.

The fit index criteria employed exhibit a high degree of leniency. Nevertheless, there has been a recent surge in the utilization of more stringent metrics. A CFI or TLI value of .90 or lower is deemed unsatisfactory. It is probable that this will give rise to challenges when attempting to replicate the factor structure in a different sample. „The results suggest that, for the ML method, a cutoff value close to .95 for TLI, BL89, CFI, RNI, and Gamma Hat; a cutoff value close to .90 for Mc; a cutoff value close to.08 for SRMR; and a cutoff value close to .06 for RMSEA are needed before we can conclude that there is a relatively good fit between the hypothesized model and the observed data. Furthermore, the 2-index presentation strategy is required to reject reasonable proportions of various types of true-population and misspecified models. Finally, using the proposed cutoff criteria, the ML-based TLI, Mc, and RMSEA tend to overreject true-population models at small sample sizes and thus are less preferable when the sample size is small." (Hu & Bentler, 1999).

In one of the more comprehensive and widely cited evaluations of cutoff criteria, the findings of simulation studies by Hu and Bentler (1999) suggest the following guidelines for acceptable model fit: (1) SRMR values close to .08 or below; (2) RMSEA values close to .06 or below; and (3) CFI and TLI values close to .95 or greater (Brown & Moore, 2012).

Given the information provided above, the TLI value of 0.889 appears to be unsatisfactory. Therefore, it would be beneficial to consider alternative models such as bifactor or estimating methods like ESEM or BESEM. Additionally, it may be beneficial to employ exploratory factor analysis (EFA) to investigate the items that are lacking in strength.

 Additionally, it would be beneficial to evaluate and demonstrate the normality of the variables, as this may also influence the method of factor analysis chosen.

 I was pleased to read your well-structured and precisely written manuscript. In my opinion, the additional notes and analysis mentioned above would greatly enhance the value of the paper.

References

Brown, T.A., Moore, M.T. (2012). Confirmatory factor analysis. Handbook of structural equation modeling. 361, 379

Hu, L. T., & Bentler, P. M. (1999). Cutoff criteria for fit indexes in covariance structure analysis: Conventional criteria versus new alternatives. Structural equation modeling: a multidisciplinary journal, 6(1), 1-55.

Author Response

Point-by-point response to Comments and Suggestions for Authors

Comments 1:  Incorporating a brief section in the introduction explaining the psychometric qualities of instruments across various countries, such as factor structures, would be beneficial.

Response 1: Thank you for pointing this out. I/We agree with these comments. Therefore, I/we have added more resources in the introduction to provide more information about the psychometric qualities of instrument across various countries. (line 126-144)

Comments 2:  The fit index criteria employed exhibit a high degree of leniency. Nevertheless, there has been a recent surge in the utilization of more stringent metrics. A CFI or TLI value of .90 or lower is deemed unsatisfactory. It is probable that this will give rise to challenges when attempting to replicate the factor structure in a different sample. „The results suggest that, for the ML method, a cutoff value close to .95 for TLI, BL89, CFI, RNI, and Gamma Hat; a cutoff value close to .90 for Mc; a cutoff value close to.08 for SRMR; and a cutoff value close to .06 for RMSEA are needed before we can conclude that there is a relatively good fit between the hypothesized model and the observed data. Furthermore, the 2-index presentation strategy is required to reject reasonable proportions of various types of true-population and misspecified models. Finally, using the proposed cutoff criteria, the ML-based TLI, Mc, and RMSEA tend to overreject true-population models at small sample sizes and thus are less preferable when the sample size is small." (Hu & Bentler, 1999).

In one of the more comprehensive and widely cited evaluations of cutoff criteria, the findings of simulation studies by Hu and Bentler (1999) suggest the following guidelines for acceptable model fit: (1) SRMR values close to .08 or below; (2) RMSEA values close to .06 or below; and (3) CFI and TLI values close to .95 or greater (Brown & Moore, 2012).

Given the information provided above, the TLI value of 0.889 appears to be unsatisfactory. Therefore, it would be beneficial to consider alternative models such as bifactor or estimating methods like ESEM or BESEM. Additionally, it may be beneficial to employ exploratory factor analysis (EFA) to investigate the items that are lacking in strength.

Response 2: Thank you for pointing these out. I/We agree with these comments that the fit index criteria exhibit a high degree of leniency. We agree that Hu & Bentler’s guidelines (1999) would be the standard criteria to follow, and we have mentioned in the study of this guideline and have included Hooper’s guidelines (2008) for interpretation of marginal values that could be considered “acceptable” fit. We are indeed in full agreement that the values could be better if alternative models were employed. However, given our time and resources, these results are the best we could produce as of now. We have mentioned these limitations in the paper and have included them as suggestions for future studies. Thank you for your comment and your suggestions.

Comments 3:  Additionally, it would be beneficial to evaluate and demonstrate the normality of the variables, as this may also influence the method of factor analysis chosen.

Response 3: As for the last comment, we have taken your suggestion and included Table 3 (line 500) to demonstrate the normality of the continuous variables of predictors and covariates.

Thank you for taking time out of your busy schedule to read the revisions.

Reviewer 3 Report

Comments and Suggestions for Authors

Thank you for the opportunity to review this manuscript. The authors report the results of a study aimed at validating the DASS-21 in the unique context of Guam, based on data collected by the PICCS study from 2017 to 2019. Additionally, they employed structural equation modeling to explore the associations between anxiety, depression, stress, and other health indicators such as diet, sleep, and physical activity.

While the study has merit and my overall impression is positive, I believe there are some issues that need addressing to enhance the manuscript's suitability for publication.

Major Points:

1. The authors report the results of their analyses twice due to a high number of missing data: once using conventional methods (listwise deletion and maximum likelihood estimation) and once using an advanced method (maximum likelihood estimation with Monte Carlo integration). While I appreciate the transparency, this choice makes it challenging for readers to follow and understand the results. I suggest explaining why the advanced method is preferable and using it exclusively to report results.

2. The Introduction section is too brief and requires improvement:

a. Stress, one of the three components of the DASS-21 scale, is not adequately addressed.

b. More information about the DASS-21 is needed, including its psychometric properties, relationships between its subscales and other constructs, and its usage in relevant studies.

c. Additional information about the health indicators (diet, sleep, physical activity) and their relationship with the DASS-21 constructs is necessary.

3. When discussing the SEM results, the authors appear to imply a causal link between the health indicators and the DASS-21 scales, as suggested by their SEM model. I’m not sure that this strong link is correct. For example, while a lack of physical activity could potentially lead to depression, it's also plausible that depression could influence individuals to reduce physical activity. The possibility of bidirectional links should be considered in the discussion of results, not just in the limitations section.

Minor Points:

Page 2, rows 51-54 “Some of the common psychological tools used world-wide include Beck Depression Inventory, Beck Anxiety Inventory, and Depression Anxiety Stress Scale-21 (DASS-21), and extensive research has been done to validate the constructs and to test the reliability of the tool in various countries.” Please add citation(s).

Page 2, row 55 “In Guam, the Pacific Island Cohort of College Student (PICCS) study have utilized”, should it be “has utilized”? Consider revising to "has utilized" for clarity. Additionally, conduct another round of proofreading for possible typos.

Clarify why diet, sleep, and physical activity were chosen as health indicators and whether they were the only options available in the PICCS study.

Page 2, rows  90-91 “This number is also well above the minimum recommended sample size (200) for testing the theoretical model of Confirmatory Factor Analysis”. Add a citation to support the claim about the recommended sample size.

Include measures of reliability (e.g., Cronbach’s alpha or McDonald omega) for other measures of interest (diet, sleep, physical activity).

Page 4, rows 140-141 “The validity of the constructs of depression, anxiety, and stress was tested first by examining the normality of the data measured”. Please clarify that the validity testing pertains to the scale, not the construct.

Page 5, row 192 and Page 7, row 243: Revise statements about model fit to reflect the actual acceptability according to cut-off values in Tables 2 and 5 (the fit is acceptable, not good).

Page 6, rows 211-214: Instead of mode values, report means and standard deviations. Consider summarizing descriptive statistics in a new table.

Page 7, row 229 “gender was positively correlated with anxiety (r = 0.149, p<.01)”. Reconsider the analysis for correlation with gender, as correlation may not be appropriate for categorical variables.

Page 7, rows 238-242: Consider reporting SEM results as Betas instead of rs.

Page 9, row 313 “leading to low reliability of the scores”. Clarify that reliability for these measures has not been reported.

Round 2

Reviewer 2 Report

Comments and Suggestions for Authors

Thank you for the corrections and I am happy to recommend the manuscript for publication. Congratulations to the authors for an interesting and high quality paper. 

Author Response

Thank you for the corrections and I am happy to recommend the manuscript for publication. Congratulations to the authors for an interesting and high quality paper. 

Author's response to reviewer's comments:

Thank you so much! Thank you for taking time out of your busy schedule to review our paper. We appreciate your time, effort, and care for quality research!

Reviewer 3 Report

Comments and Suggestions for Authors

I appreciate the authors' efforts in addressing my feedback, resulting in notable improvements, particularly in the Introduction and results sections. However, there remain unresolved data analysis issues critical for publication consideration:

- Reliability of the Diet questionnaire, essential for psychometric assessment, remains unreported despite the authors' claim of time and resource constraints, which is unacceptable in contemporary research standards.

- The assertion of the Diet questionnaire being a categorical variable contradicts the continuous nature described in the manuscript (page 5, rows 201-204, “The scores for the frequency of each food type were summed up and ranged from 0-9, where 9 indicated a high frequency of fruit and vegetable consumption and a low frequency of fast-food consumption.”). Mean and standard deviation reporting are warranted.

- Despite claims of correction, gender, a categorical variable, is still treated as continuous in correlation analysis and Table 3, necessitating rectification.

- Inconsistent representation of SEM results as rs in the text and standardized estimates in Table 5 requires alignment, potentially by renaming text values to betas for clarity.
